# Renoprotective Effects of Mangiferin: Pharmacological Advances and Future Perspectives

**DOI:** 10.3390/ijerph19031864

**Published:** 2022-02-07

**Authors:** Sumaya Akter, Akhi Moni, Golam Mahbub Faisal, Muhammad Ramiz Uddin, Nourin Jahan, Md Abdul Hannan, Asadur Rahman, Md Jamal Uddin

**Affiliations:** 1ABEx Bio-Research Center, East Azampur, Dhaka 1230, Bangladesh; sumayabge.nstu@gmail.com (S.A.); akhimoni840818@gmail.com (A.M.); golammahbub00@gmail.com (G.M.F.); muhammadramizuddin@gmail.com (M.R.U.); hannanbmb@bau.edu.bd (M.A.H.); 2Faculty of Veterinary Medicine and Animal Science, Bangabandhu Sheikh Mujibur Rahman Agricultural University, Gazipur 1706, Bangladesh; 3Department of Pharmacology, Faculty of Medicine, Kagawa University, 1750-1 Ikenobe, Miki-cho, Kita-gun, Takamatsu 761-0793, Japan; s19d724@stu.kagawa-u.ac.jp; 4Department of Biochemistry and Molecular Biology, Bangladesh Agricultural University, Mymensingh 2202, Bangladesh; 5Graduate School of Pharmaceutical Sciences, College of Pharmacy, Ewha Womans University, Seoul 120-750, Korea

**Keywords:** chronic kidney disease, renoprotective, kidney fibrosis, oxidative stress, inflammation, mangiferin

## Abstract

Both acute and chronic kidney diseases substantially contribute to the morbidities and mortality of patients worldwide. The existing therapeutics, which are mostly developed from synthetic sources, present some unexpected effects in patients, provoking researchers to explore potential novel alternatives. Natural products that have protective effects against various renal pathologies could be potential drug candidates for kidney diseases. Mangiferin is a natural polyphenol predominantly isolated from *Mangifera indica* and possesses multiple health benefits against various human ailments, including kidney disease. The main objective of this review is to update the renoprotective potentials of mangiferin with underlying molecular pharmacology and to highlight the recent development of mangiferin-based therapeutics toward kidney problems. Literature published over the past decade suggests that treatment with mangiferin attenuates renal inflammation and oxidative stress, improves interstitial fibrosis and renal dysfunction, and ameliorates structural alteration in the kidney. Therefore, mangiferin could be used as a multi-target therapeutic candidate to treat renal diseases. Although mangiferin-loaded nanoparticles have shown therapeutic promise against various human diseases, there is limited information on the targeted delivery of mangiferin in the kidney. Further research is required to gain insight into the molecular pharmacology of mangiferin targeting kidney diseases and translate the preclinical results into clinical use.

## 1. Introduction

Kidney disease is a significant public health problem affecting more than 750 million people worldwide [1]. The burden of kidney disease varies substantially across the world, as does its detection and treatment. Kidney diseases are broadly classified into acute kidney injury (AKI) and chronic kidney disease (CKD). AKI denotes spontaneous kidney damage that generally lasts a few days to a week. The leading causes of AKI are damage to the kidney tissue by drugs or infections, leading to the blockage of urine (for example, a blockage can also be caused by kidney stones) [2]. If these conditions persist, kidney function decreases over time, leading to CKD. In the worst-case scenario, end-stage renal disease (ESRD), also known as kidney failure, can develop. There are many other factors or conditions involved in CKD, such as diabetes mellitus, glomerulonephritis, nephrosclerosis, and polycystic kidney disease [2]. There are many different causes of kidney disease, and sometimes the cause is unknown.

Renoprotective effects imply preserving the kidney structure or function in different pathological conditions. To date, there is no specific treatment option to completely inhibit the progression of AKI and CKD by preserving the integrity of architecture or function of kidneys. However, therapies in terms of kidney protection to confront the risk factors are worthy of consideration for maintaining the patient’s kidney function. A plethora of traditional treatments using natural products have shown a wide range of therapeutic windows for kidney diseases. Among these, one of the exemplary candidates is mangiferin (molecular formula: C19H18O11; systematic name: 1,3,6,7-tetrahydroxyxanthone C2-β-D-glucoside) (Figure 1), a naturally occurring glucoxilxanthone [3,4] derived from the various parts of *Mangifera indica* (Mango), including the leaves, fruits, flowers, seeds, roots, and stem bark [5]. In China, mangiferin is included in many traditional formulae that contain Iris domestica, Folium pyrrosiae, Gentiana scabra, and A nemarrhena asphodeloides. Moreover, this natural bioactive and polyhydroxy polyphenol element is enriched with several pharmacological beneficial effects without any known side effects [6].

Mangiferin is shown to have a renoprotective effect. Therefore, it has been extensively investigated regarding the beneficial effects of kidney diseases. Several reports support that mangiferin confers its renoprotective effects against AKI and CKD predominantly through protection against inflammation, scavenging of reactive oxygen species (ROS) in oxidative stress, anti-apoptotic and anti-fibrotic effects in renal intestinal fibrosis, preserving mitochondrial function, and reducing lipid peroxidation [7,8]. An overview of various studies on mangiferin against kidney diseases is presented in this review to gain insight into the potential pharmacological roles of mangiferin in kidney diseases. We attempt to outline the study design of each relevant study on kidney diseases in terms of the animal or cell culture model, mangiferin’s dose, and molecular results. A summary of the collection of information on the effectiveness of mangiferin in kidney diseases is then presented. The possible protective mechanisms of mangiferin are also highlighted and discussed to fill the knowledge gap regarding its application as an alternative treatment for kidney diseases.

## 2. Search Strategy

The literature was collected from online research databases such as PubMed, and Google Scholar using the keywords ‘mangiferin on kidney diseases’ and ‘mangiferin on oxidative stress, inflammation, and fibrosis in kidney diseases’. Thereafter, the sorted in vivo and in vitro findings regarding the renoprotective effects of mangiferin from 2002 to 2021 were summarized in this review. All figures were generated using adobe illustrator.

## 3. Mangiferin: Natural Sources and Extraction Methods

Mangiferin is predominantly isolated from *Mangifera indica* (mango). Various parts of this plant, including leaves, stem bark, fruit peels, kernel, and root were subjected to isolation of mangiferin. Mangiferin could be extracted using several techniques, including maceration-assisted extraction, reflux extraction, solid-phase microextraction, and hydro distillation; however, all of these conventional extraction techniques are not cost-effective and eco-friendly in terms of solvent consumption and time requirement [9,10]. In recent years, some novel extraction methods have been established which are rapid, less time-consuming, efficient, and eco-friendly. The most used techniques include microwave-assisted extraction (MAE), ultrasonic extraction, supercritical fluid extraction, enzymatic extraction, and dispersive liquid-liquid microextraction [10,11].

Lerma-Torres and co-researchers demonstrated that sonication resulted in the highest yield of mangiferin (1.45 g 100 g^−1^ dried bark) compared to maceration, Soxhlet, and microwave-assisted extractions, indicating that ultrasound-assisted extraction could be an effective alternative to conventional extraction techniques [12]. In another study, the highest mangiferin content was reported in the peel of Lvpimang variety of mango fruit (7.49 mg/g DW) using a combination of macroporous HPD100 resin chromatography with optimized high-speed counter-current chromatography (HSCCC) [13].

To maximize the extraction of mangiferin from *M. indica* leaves, Kulkarni and Rathod used a combined technique named three phase partitioning (TPP) coupled with ultrasound (UTPP) [14]. At the optimized conditions of UTPP (time 25 min, pH 6, ammonium sulfate saturation 40% *w*/*v*, slurry to t-butanol ratio 1:1, solute to solvent ratio 1:40, frequency 25 kHz, power 180 W, duty cycle 50%, soaking time 5 min and temperature 30 ± 2 °C), the mangiferin yield was 41 mg/g in 25 min which was ~1.5 times higher than TPP (28 mg/g in 2 h). In a prelusive scale extraction of mangiferin from *M. indica* leaves, Anbalagan and the team investigated the effect of solvent type, leaf age, extraction time, and extraction temperature (40–70 °C) on the recovery of mangiferin [15]. At the optimized conditions (extraction time 6 h, temperature 70 °C, and sample mass to solvent volume ratio 1:15), extraction with ethanol gave the highest yield compared to that with acetone, ethyl acetate, and hexane. As estimated at the same conditions, young leaves (22.28%) contain a higher amount of mangiferin than old leaves (10.74%).

Several other groups reported mangiferin extraction from other plant sources. Alara and the team extracted mangiferin from *Phaleria macrocarpa* fruits using response surface methodology [16]. Chavan reported the highest mangiferin content in callus cultures of *Salacia chinensis* L. enriched by jasmonic acid treatment [17]. Aquilaria leaf was subjected to mangiferin extraction using ultrahigh-performance liquid chromatography (UHPLC) coupled with electrospray ionization (ESI) tandem mass spectrometry (MS/MS) method [18]. Mangiferin was also extracted from *Aphloia theiformis* using liquid-liquid extraction followed by UPLC–QTOF–MS [19].

## 4. Pharmacological Effects of Mangiferin on Kidney Diseases

The pharmacological potential of mangiferin against several plausible factors such as oxidative stress, inflammation, fibrosis, and other pathologies associated with kidney diseases are summarized in this section (Figure 2, Figure 3 and Figure 4 and Table 1 and Table 2).

### 4.1. Effects of Mangiferin on Renal Oxidative Stress

Oxidative stress arises in the cells owing to an imbalance between production and accumulation of ROS, and subsequent inability to detoxify these reactive products (Figure 2) [40,41,42]. ROS is generated in the cells during the normal process of metabolism. Although optimum ROS level helps cell metabolism by cell-to-cell interactions, elevated ROS can cause extreme damage to cells and tissues [43]. The antioxidative defense mechanism of cells combined with an exogenous source of antioxidants is the key to minimizing the damages occurred by oxidative stress. Mangiferin, the natural bioactive compound has already been proven to have antioxidant properties. It may maintain a proper balance between ROS and antioxidants by reducing the level of intracellular ROS. It increases antioxidant activities in the kidney tissues, and subsequently stabilizes superoxide dismutase (SOD) activities, decreases uric acid synthesis, and improves antioxidant effects in mice and rats (Table 1).

Cisplatin (cis-diamminedichloroplatinum II), is an anticancer drug that acts against several types of cancers by increasing intracellular ROS levels leading to oxidative stress, and reducing the antioxidant enzyme activities in the kidney epithelial (NKE) cells [20]. Importantly, mangiferin treatment reduces the kidney ROS levels, oxidative stress marker specifically MDA, and enhanced the antioxidant enzyme activities namely SOD, catalase, glutathione s-transferase (GST), glutathione peroxidase (GPx), and glutathione reductase (GR) are responsible for the degradation of ROS in the renal tissues both in vitro (NKE cells) [20] and in vivo (mice and rats) [20,21].

Several other drugs/chemical compounds cause oxidative damage by increasing ROS levels. Streptozotocin (STZ), a well-known poisonous substance forms ROS and MDA, resulting in induction of oxidative stress and reducing the levels of antioxidants [44]. Interestingly, mangiferin increased antioxidant enzymes as well as reduced ROS and MDA in STZ-induced diabetic mice and rats [7,22,23,24,25,26,27]. In diabetes, advance glycation end product (AGE) and xanthin oxidase plays an important role in inducing ROS; however, mangiferin treatment suppressed AGE generation and inhibited the activities of xanthin oxidase [22], indicating that mangiferin has the potential to reduce the levels of ROS in diabetic patients.

Mangiferin reduces the damage of antioxidants in lipopolysaccharide (LPS)-induced sepsis, thus, recovering sepsis-associated organ impairment [28] and increases antioxidant biomarkers in mouse kidney cells [29]. Besides, it has also been reported that mangiferin reduces intracellular MDA levels by acting as an effective ROS scavenger. Consistently, mangiferin ameliorated the oxidative stress in animal models with different ROS inducers namely uric acid, osteopontin, tert-Butylhydroperoxide (tBHP) and D(+)galactosamine (DGal) [30,31,32,33,34,35] as well as in different kidney cells [37,38,39], as briefly described in Table 1 and Table 2. Collectively, all these data from in vivo and in vitro studies indicated that mangiferin could be a potent therapeutic agent to combat oxidative stress in kidney diseases.

### 4.2. Effects of Mangiferin on Renal Inflammation

Renal inflammation is initiated by several factors including immune-mediated inflammatory mediators and a subsequent renal dysfunction or nephrotoxicity (Figure 3) [45,46,47,48,49]. Moreover, increased uric acid triggers renal inflammation through the c-Jun N-terminal kinases (JNK) signaling pathway and nucleotide-binding oligomerization domain (NOD)-, leucine-rich repeat (LRR)- and pyrin domain-containing protein (NLRP) 3 inflammasome. NLRP3 inflammasome stimulates pro-inflammatory cytokines interleukin-1β (IL-1β) and IL-18 production and excretion, thus promoting renal injuries and disease in mice such as hyperuricemic nephropathy (HN) [30]. In connection with these, recent studies demonstrated the anti-inflammatory effects of mangiferin in kidney tissues and cells (Table 1 and Table 2). Mechanistically, mangiferin shows its anti-inflammatory effects by inhibiting the activation of both the JNK pathway and NLRP3 inflammasome and minimizing urate [29]. It has also demonstrated that mangiferin treatment preserved the glomerular and tubular structures of kidneys in mice [30]. Cisplatin manifests nephrotoxicity in the liver with deleterious effects in the renal tissues. Cisplatin exposure escalates proinflammatory cytokines (tumor necrosis factor-α; TNF-α, IL-1β, IL-6, IL-10) and nuclear factor-kappa β (NF-κB) in the nuclear fraction of the renal tissue in rats and mice; however, mangiferin treatment attenuated the level of all these cytokines [20,21].

Studies demonstrated that high mobility group box 1 (HMGB1), a protein that binds to toll-like receptors (TLR), activates NF-κB inflammatory signaling pathways inducing the release of inflammatory factors; hence, it contributes to an inflammatory storm in LPS-treated mice, and interestingly, mangiferin prevented the activation of this signaling pathway [28]. Moreover, simultaneous administration of mangiferin reduced cisplatin-instigated nephrotoxicity in vitro and in vivo by preventing the nuclear translocation of NF-κB and proinflammatory cytokines. Blocking NF-κB pathway by mangiferin defeats the NF-κB cascade pathway of inflammation in rats and mice [20,21]. In contrast, tBHP instigated inflammation [34], as well as STZ induced-diabetic nephropathy (DN) [7,22,23,24,25,26,27] were ameliorated by mangiferin. Moreover, DGal advances inflammation by prompting NF-κB and TNFα in rats in renal tissue, and osteopontin (OPN), a proinflammatory cytokine, assists in inflammatory gene expression. Mangiferin preserves renal function appositely against NLRP3 inflammasome [33] and lessens inflammation by obstructing DGal [35] and OPN [33]. In diabetic nephropathy, high glucose generates an inflammatory response characterized by activating the NF-κB pathway, TNF-α, and IL-1β in the renal tissue of diabetic rats, and mangiferin was able to reduce the burden of inflammation in DN [39]. Moreover, mangiferin inhibits renal ischemia-reperfusion damages by blocking inflammatory agents [36] and preserves renal function against cadmium-induced eukaryotic cell death via the NF-κB signaling pathways [38]. All data suggest that mangiferin has the ability to combat inflammation in the kidneys.

### 4.3. Effects of Mangiferin on Renal Fibrosis

Persistent accumulation of extracellular matrix (ECM) in the renal tissue causes kidney fibrosis. ROS helps to intensify the accumulation of ECM by inducing transforming growth factor (TGF)-β1, which is evident in the in vivo mice models [7,50,51,52]. Cellular ROS also leads to activate fibrotic factors such as fibronectin (FN), α-smooth muscle actin (α-SMA), and collagen I (Col I) (Figure 4). FN is a non-collagenous glycoprotein present in ECM and basement membrane, which helps cell adhesion and regulates cell polarity, differentiation, and enlargement, and mangiferin prevents to activate FN [7]. The active protein kinase C beta (PKCβ) pathway contributes to renal fibrosis in hyperuricemia boosted mice [30]. PKC is a family of enzyme threonine kinases along with 12 isoforms. Activation of PKC isoforms enhances the vasoendothelial growth factor (VEGF), leading to Col IV deposition and mesangial growth in rats [23]. OPN is known as ECM protein, and pro-fibrotic adhesion compound induces renal disease tubulointerstitial fibrosis. OPN can act as a mediator for fibrotic changes related to glomerulosclerosis and interstitial fibrosis, however, mangiferin treatment attenuated the development of these pathological changes in kidneys [33].

Mangiferin may act as a new therapeutic agent against chronic fibrotic kidney diseases such as HN and DN [32]. Treatment with mangiferin impedes the expression of FN in hyperuricemic nephropathy in mice [30]. Evidence shows that mangiferin administration effectively shows anti-fibrotic effects against renal interstitial fibrosis by decreasing TGF-β1-induced elevation of Col I, FN, and α-SMA and by inhibiting glomerular ECM development, accumulation, reducing glomerular basement membrane thickness, as well as mesangial cell growth in STZ, instigated diabetic mice [7]. Taken together, mangiferin could become a potential therapeutic option to inhibit the fibrotic changes in kidneys.

### 4.4. Effects of Mangiferin against Other Kidney Pathologies 

ROS activates mitogen-activated protein kinases (MAPK) pathways along with JNK, p38 kinases, and extracellular signal-regulated kinases (ERK) by inducing cellular stress in rats [7]. The downstream pathway of MAPK involves three essential serine kinase proteins, including JNK, ERK1/2, and p38. These proteins assist in apoptosis and cell death. Importantly, mangiferin treatment reduced ROS generation and subsequently alter the MAPK pathway [21]. In contrast, proapoptotic (Bax, Bad, etc.) and anti-apoptotic (Bcl-2, Bcl-xl) proteins are entangled in apoptosis in STZ persuaded diabetic rats, which is attenuated by mangiferin treatment [22]. TNF-α activates caspase 8, while caspase 8 and 3 play a significant role to ensure programmed cell death that is apoptosis. Also, phosphorylation of JNK protein executes apoptosis and mangiferin treatment successfully prevents apoptosis in kidney tissues [22]. DGal, a toxic component, upregulates caspase 3/9 and changes the reciprocal regulation of Bcl-2 family proteins and these proteins govern permeabilization of outer membrane mitochondria. Bax and Bad proteins initiate apoptosis, release cytochrome c into the cytosol, and gradually activate the pro-apoptotic caspase cascade. Anti-apoptotic proteins Bcl-2 and Bcl-xL suppress Bax and Bak oligomer [35]. AMP-activated protein kinase (AMPK), a serine protein kinase that responds to cellular stress is activated when cellular energy is insufficient, thus inhibits mammalian target of rapamycin complex (mTOR) activity and increases autophagy. Autophagy refers to the degradation associated with the clearance of injured proteins and organelles initiated by activating the unc-51-like kinase 1 (ULK1) complex. ULK1 complex is prevented by mTOR activities AMPK phosphorylation decreases mTOR phosphorylation and manifests a nephroprotective effect by regulating MAPK. Mangiferin treatment was proven to have beneficial effects on these molecular machineries [24]. Moreover, pre-treatment with mangiferin exerts uricosuric action in hyperuricemic rats associated with obstruction of urate reabsorption through down-regulation of the mRNA and protein expressions of urate transporters in renal cells [32]. It also damages apoptotic proteins such as p53 and Nrf-2 related signaling cascades, reduces caspase and mitochondrial dysfunction [20,22]. Furthermore, it rehabilitates the altered Bax/Bcl-2 ratio by reducing mitochondrial dysfunction, releasing cytochrome C in the cytoplasm from the mitochondria [30], and diminishing Bax proteins in DGal revealed rats [35]. Mangiferin exerts beneficial effects on these different molecular cascades.

A mangiferin aglycon derivative J99745 has been used as a potent xanthine oxidase (XOD) inhibitor in vitro study. J99745 exerts urate-lowering effect by inhibiting XOD activity and renal urate transporter 1 (URAT1) expression [31].

## 5. Biosynthesis and Bioavailability of Mangiferin

Although mangiferin acts as a therapeutic agent against renal diseases, low bioavailability, shorter half-life, poor solubility, and quick removal from the body minimize the efficacy of mangiferin and restrict to act as a potent therapeutic [53]. Mangiferin nanoparticles were recently improved to enhance the solubility and bioavailability of mangiferin. Hydrogenated soy phosphatidylcholine (HSPC), a phospholipid carrier system for phytomolecules, assists in medication delivery. Bhattacharyya et al. [53] demonstrated that complexation with phospholipid improves the bioavailability and bioactivity of these phytochemicals. Aerial parts and rhizomes of Anemarrhena asphodeloides contain mangiferin and isomangiferin. Inoue and Fujita [54] demonstrated mangiferin and its isoform isomangiferin biosynthesis in anemarrhena asphodeloides through a retrobiosynthetic process. Biosynthesis of mangiferin is also related to flavonoids. Studies explained that extracted benzophenone C-glycosyltransferase from *M. indica* (MiCGT) effectively caused C-glucosylation, which might be an alternative approach for the biosynthesis of mangiferin [55]. Therefore, to improve the bioavailability of mangiferin, future investigations would be important to integrate some structural change to establish it as a potent therapeutic option, especially for kidney diseases.

## 6. Pharmacological Advances of Mangiferin-Based Drug Development

As mangiferin has the potential to be a drug candidate against kidney diseases, a great number of studies have been carried out to establish a relationship among multi-potent drugs based on mangiferin and to characterize their potential targets. Mangiferin shows limited bioavailability due to its poor solubility. Various nanoparticles incorporating mangiferin have been developed to improve the solubility and bioavailability of mangiferin. Nanoparticles are enriched with free radical scavenging activity, lipid peroxidation, protein oxidation inhibition activity, and synergistic action with phase II antioxidant enzymes such as catalases and peroxidases [37]. Phospholipid complex-loaded self-assembled phytosomal soft nanoparticles incorporating mangiferin showed improved solubility, ex vivo permeability, oral bioavailability, and antioxidant potential of mangiferin [56]. Pre-treatment with the mangiferin-chitosan nanoparticles (MCNs) exclusively prevents the induction of NaF-induced cytotoxicity and maintains the level of intracellular antioxidant enzymes in NKE cells. These nanoparticles can be used in the food and pharmaceutical industries as a therapeutic agent to prevent oxidative stress-induced kidney disorders [37]. No reported toxicity of mangiferin conjugated gold nanoparticles (AuNP) also suggested their future use as a drug delivery system and other related medicinal uses in kidney disease [57]. In addition, mangiferin-loaded nanoparticles have been shown potential against various other conditions. Mangiferin-alginate grafted N-succinylated chitosan (NSC) nanoconjugate was shown to lower blood glucose, total cholesterol, and triglycerides in diabetes mediated hyperlipidemic in rats [58]. Mangiferin functionalized gold nanoparticles (MGF-AuNPs) showed immunomodulatory intervention against prostate cancers [59]. Chitosan-silica hybrid scaffolds loaded with mangiferin have shown promise in bone regeneration [60]. Also, the use of mangiferin-integrated polymer systems as a new potential anticancer agent has been reviewed [61]. All these pieces of evidence indicate the therapeutic potential of mangiferin-loaded nanoparticles in various diseases, including kidney diseases (Table 3).

## 7. Conclusions and Future Directions

With the increasing incidence of kidney complications and the limitation of existing therapeutics, researchers continue to explore alternative therapies. Various natural products having renoprotective effects can be a prospective alternative. The contributions to the literature highlighted in this review have suggested that mangiferin is effective in attenuating kidney inflammation, improving renal fibrosis, and repairing tissue damage caused by various toxins, drugs, and infections. The anti-inflammatory, antioxidant, antiapoptotic, and antifibrotic effects are mainly attributed to the renoprotective effects of mangiferin. This information further suggests that mangiferin could be developed as a promising therapeutic agent for kidney diseases.

Although renoprotective potentials of mangiferin have been supported by a substantial number of experimental studies, this evidence is mostly from preclinical studies and therefore human trials are warranted to confirm its clinical use. Since the clinical use of mangiferin is limited due to its poor bioavailability, approaches such as modification of its structure without hindering biological activity may improve the bioavailability. Strategies, including nanoparticle-guided target delivery, are also currently under investigation to improve bioavailability. Furthermore, the safety of using mangiferin as a therapeutic agent has not been unequivocally established. Integrated experimental and bioinformatics approaches are essential to address all the aforementioned issues. The prospects and limitations of the clinical application of mangiferin highlighted in this review will surely inspire future research to develop mangiferin-based therapeutic agents for patients with kidney diseases.

## Figures and Tables

**Figure 1 ijerph-19-01864-f001:**
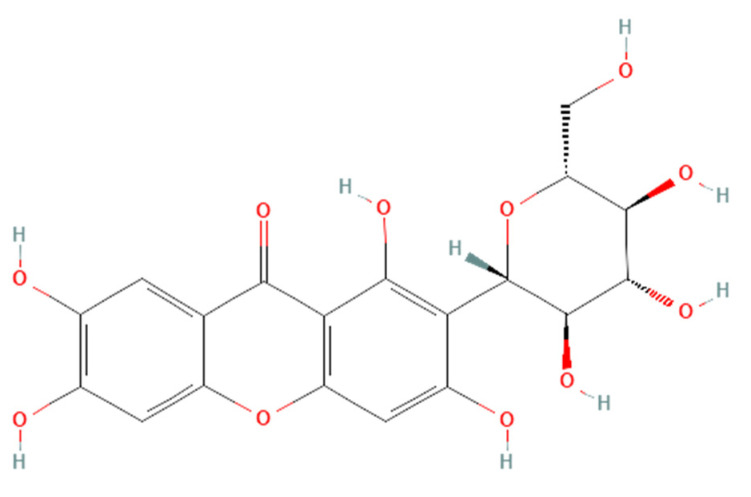
Structure of mangiferin (PubChem CID 5281647).

**Figure 2 ijerph-19-01864-f002:**
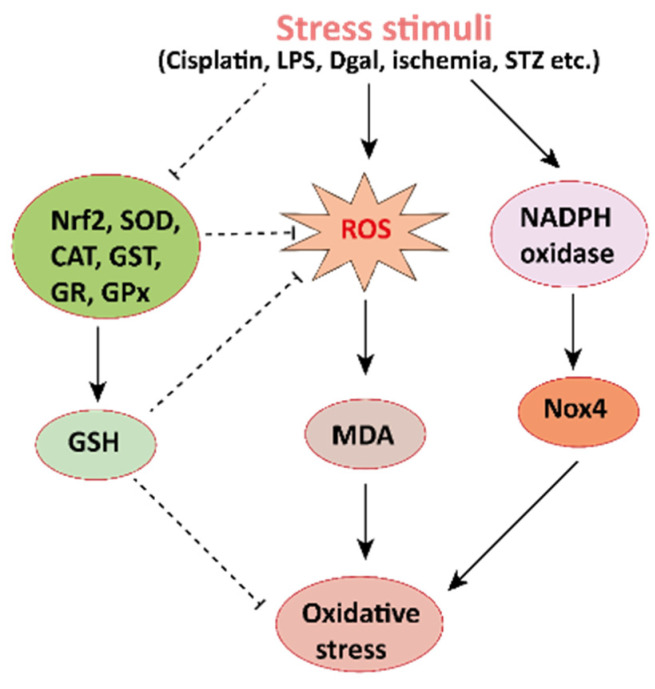
Mechanisms involved in the pathogenesis of oxidative stress in the kidney. NADPH oxidase is the main source of cellular ROS. Nox4 is an isoform of NADPH oxidase expressed in renal tubules that leads to oxidative stress (ROS and MDA) and damages the kidney. Stress stimuli, for instances cisplatin, STZ, ischemia result in decreased Nrf2 thus leading to oxidative stress. Reactive oxygen species, ROS; MDA, malondialdehyde; streptozotocin, STZ; NF-E2-related factor 2, Nrf2.

**Figure 3 ijerph-19-01864-f003:**
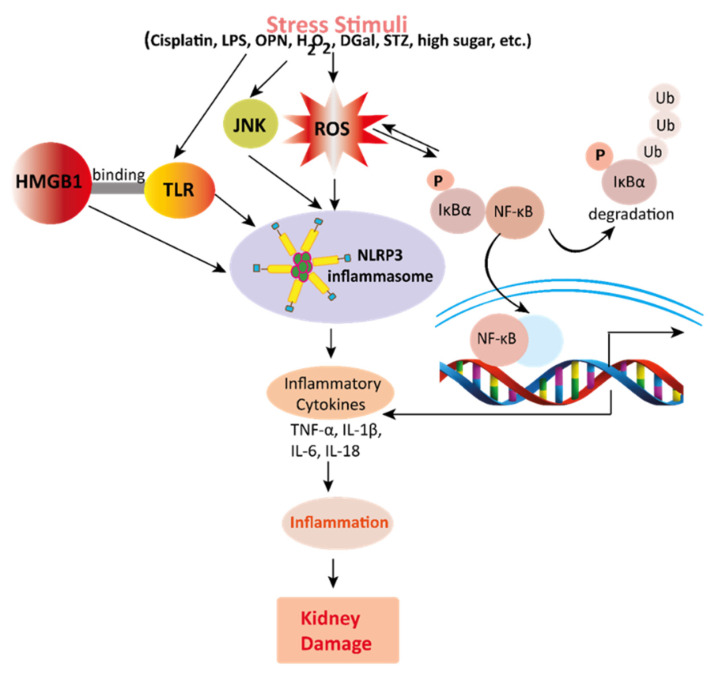
Mechanisms involved in the pathogenesis of inflammation in the kidney. Stress stimuli example for, cisplatin, DGal, STZ, LPS, OPN, high glucose, and H_2_O_2_ generate ROS in the kidney. Accumulation of ROS induces inflammation through the activation of the NLRP3 inflammasome. NLRP3 involves a multi-protein complex known as inflammasome, which triggers the NF-κB signaling pathway. HMGB1 protein is a late inflammation instigating compound which activates the NF-κB signaling pathway by binding with Toll-like receptors. The NF-κB pathway further promotes an inflammatory storm by releasing inflammatory cytokines (TNF-α, IL-1β, IL-6, IL-18), which ultimately leads to kidney damage. High-mobility group box 1; HMGB1, D (+) galactosamine; DGal, osteopontin; OPN, Streptozotocin; STZ, reactive oxygen species; ROS, IL-6; interleukin-6, IL-1β; Interleukin 1β, TNFα; Tumor Necrosis Factor-α.

**Figure 4 ijerph-19-01864-f004:**
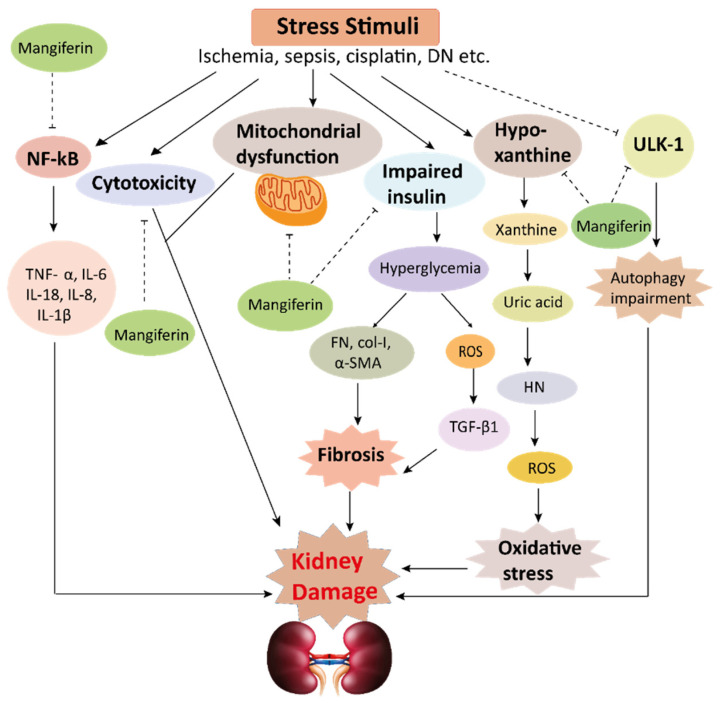
This schematic representation shows that stress stimuli example for, cisplatin, DN, Ischemia, and sepsis mediates various pathological conditions including cytotoxicity, oxidative stress, inflammation, fibrosis, autophagy dysfunction, and mitochondrial dysfunction. These ultimately lead to kidney damage. Stress stimuli activate the NF-kB signaling pathway which triggers the release of inflammatory cytokines (TNF-α, IL-6, IL-18, IL-8, and IL-1β) and decreases the action of ULK-1 thus autophagy impairment causes. Further col-1, FN, α-SMA causes accumulation of extracellular matrix (ECM) resulting in fibrosis. Mangiferin protects the kidney by suppressing the cascades of inflammatory pathways, oxidative stress, fibrosis, cytotoxicity, mitochondrial dysfunction, and autophagy impairment. ROS, reactive oxygen species; TNF-α, tumor necrosis factor-α; IL-1β, interleukin-1β; FN, fibronectin; α-SMA, α-smooth muscle actin; Col I, collagen I; TGF-β1, transforming growth factor, ULK-51, unc-51-like kinase; DN, diabetic nephropathy; HN, hyperuricemic nephropathy; NF-kB, nuclear factor-kappa B.

**Table 1 ijerph-19-01864-t001:** In vivo renoprotective effects of mangiferin.

Model Animals	Disease Inducing Agents	Mangiferin Dosages			Effects of Mangiferin				Ref.
		Dose	Route	Duration	Oxidative Stress	Inflammation	Fibrosis	Other Pathologies	
Mice	Cisplatin	10, 20 and 40 mg/kg	orally	21 days	↓ ROS ↑ GSH, SOD, CAT, GST, GR, GPx	↓ TNF-α, IL-1β, IL-6, IL-10, NF-κB	−	↓ Caspase-3	[20]
Rats	Cisplatin	20 and 40 mg/kg	i.p.	10 days	↓ ROS, MDA ↑ GSH, SOD, CAT	↓ TNF-α, IL-6	−	↓Bax, caspase-3 ↑ Bcl-2↓ MAPK pathway	[21]
Mice	STZ	15, 30 and 60 mg/kg/day	orally	4 Weeks	↓ ROS, MDA ↑ SOD, CAT, andGSH-Px	↓TNF-α, IL-6, IL-1β	↓TGF-β1, FN, Col I, and α-SMA	↓Phosphorylation of PI3K and Akt	[7]
Rats	STZ	10, 20, 40, 60 and 80 mg/kg;	orally	30 days	↓ ROS, MDA ↑ SOD, CAT, GPX, GR	↓ NF-kB, TNF-α	−	↓ Cytochrome C ↓ Bax, Caspase-9, Caspase-3↑ Bcl-2↓ MAPK pathway	[22]
Rats	STZ	40 mg/kg	orally	28 days	↓ ROS ↑ CAT, GST, GS, GSH, GPx, SOD	↓ NF-κB, TNF-α, VEGF, PKC	↓ TGFβ1	−	[23]
Rats	STZ	12.5, 25, or 50 mg/kg	orally	12 weeks	−	−	−	↑ AMPK ↓ mTOR ↑ pULK1	[24]
Rats	STZ	40 mg/kg/day	orally	30 days	↑ SOD, CAT, GPx and GSH ↓ ROS, MDA	−	−	−	[25]
Rats	STZ	10 and 20 mg/kg	i.p	28 days	↓ ROS, MDA ↑ SOD, CAT	−	−	−	[26]
Rats	STZ	15, 30, and 60 mg/kg	orally	9 weeks	↓ ROS, MDA, AGEs ↑ GSH	−	−	−	[27]
Mice	LPS	20, 50, and 100 mg/kg	i.p.	7 days	↓ ROS, MDA ↑ SOD	↓ NF-κB, HMGB1	−	−	[28]
Rats	LPS	10, 50, 100 μM		1 h	↓ ROS, MDA	↓ IL-1β, IL-18 and NLRP3	−	↓ caspase-9 and caspase-3	[29]
Mice	Uric acid	50 mg/kg/day	orally	7 days	↓ ROS, XO ↑ SOD	↓ IL-1b, IL-18	−	−	[30]
Mice	Uric Acid	10, and 30 mg/kg		7 days	↓ ROS, MDA	−	−	−	[31]
Rats and mice	Uric acid	1.5–6.0 mg/kg	i.p.	5 days	−	−	−	↓ URAT1, OAT10, and GLUT9	[32]
Rats	OPN	15, 30, and 60 mg/kg/day	orally	9 weeks	−	↓ TNF-α, COX-2, IL-1β, NF-κB, p65	↓ Col IV, α-SMA	−	[33]
Mice	tBHP	75 mg/kg	orally	2 weeks	↑ SOD, CAT, GST, GRGPX	↓ TNF-α, IL-6 and IL-1β	−	↓ Bax, caspase-8 caspase-3 ↑ Bcl-2	[34]
Rats	DGal	25 mg/kg	i.p.	14 days	↓ ROS, MDA ↑ CAT, GST, GS, GPx, SOD	↓ NF-κB a, NO, TNF-α	−	↓ Bax, cytochrome c, caspase-9 and caspase-3 ↑ Bcl-2, Bcl-xL	[35]
Mice	Ischemia	10, 30, and 100 mg/kg	i.p.	30 min	*−*	↓ TNF-α and IL-1β	*−*	↓ Caspase-3	[36]

ROS, reactive oxygen species; SOD, superoxide dismutase; GSH, glutathione; CAT, catalase; GST, glutathione S-transferase; GR, glutathione reductase; GPx, glutathione peroxidase; AGEs, advanced glycation end products; α-SMA, α-smooth muscle actin; DGal, D+galactosamine; ERK, extracellular signal-regulated kinases; FN, fibronectin; GLUT9, glucose transporter 9; HMGB1, high-mobility group box 1; i.p, intraperitoneal injection; IL-1β, interleukin 1β; IL-6, interleukin-6; JNK, c-Jun N-terminal kinases; LPS, lipopolysaccharide; MAPK, mitogen-activated protein kinases; MDA, malondialdehyde; mTOR, mammalian target of rapamycin; OAT10, organic anion transporter 10; OPN, osteopontin; PI3K, phosphoinositide 3-kinase, TNFα; tumor necrosis factor-α; TGF-β1, transforming growth factor-β1, tBHP; tert-Butylhydroperoxide, ULK1; unc-51-like kinase 1, URAT1; urate-anion transporter 1, VEGF; Vascular endothelial growth factor, XOD; Xanthone Oxidase.

**Table 2 ijerph-19-01864-t002:** In vitro renoprotective effects of mangiferin.

Cell Lines	Model Drug	Mangiferin		Effects of Mangiferin			Ref.
		Dose	Duration	Oxidative Stress	Inflammation	Other Pathologies	
NKE cells	Cisplatin	5–30 µM	2 h	↓ ROS ↑ GSH, SOD, CAT, GST, GR, GPx	−	↓ Caspase-3	[20]
NKE Cells	NaF	100–1000 μg/mL		↓ ROS ↑ CAT, peroxidase and GHS	−	−	[37]
HRGEC	Cadmium	75 μM	24 h	↓ ROS, MDA ↑ SOD, GSH, GR, GSH-Px	↓ NF-kB, IL-6, IL-8	↓ Bax, Cytochrome C, Caspase ↑ Bcl-2	[38]
Mesangial Cells (SV40 MES 13)	High glucose (25 mM)	50 mg/kg	48 hr	↓ ROS ↓ NOX4	−	↓ Caspase-3 ↑ Mitochondrial membranepotential	[39]

NKE cells, normal kidney epithelial cells; ROS, reactive oxygen species; GSH, glutathione; SOD, superoxide dismutase; CAT, catalase; GST, glutathione S-transferase; GR, glutathione reductase; GPx, glutathione peroxidase; NaF, Sodium fluoride, HRGEC, human renal glomerulus endothelial cells; MDA, malondialdehyde, IL-6, interleukin-6; IL-1β, interleukin 1β; NOX4, NADPH oxidase 4.

**Table 3 ijerph-19-01864-t003:** Effect of nanoparticle based-mangiferin treatment on various pathological conditions.

Nano-Particles	Cells/Tissues/Others	Method of Preparation	Size	Zeta Potential	Dose	Experimental Models	Indication	Benefits	Ref.
MCNs	NKE cells	Ionic gelation method	<80 nm	+25 ± 0.2 mV	100–1000 μg/mL	Nephropathic model	Nephro-protection	Decreases oxidative stress Enhances anti-oxidant activity, solubility and bioavailability	[37]
MGF treated-ZnO	Bone tissue	Sol-gel synthesis and freeze-drying	25–60 nm				Bone substitute material	Increases bone regeneration	[60]
MGF-AuNP	Prostate cell	Clathrin-mediated pathway	35 ± 2 nm	−40 ± 2 mv	41 µM	Xenograft model	Prostate protection	Reduces pro-tumor cytokines	[59]
MPLC SNPs	Hepatic tissue	Solvent evaporation and nanoprecipitation	1 nm to 10 μm	−200 to + 200 mV	100 μg/mL	Albino rat model	Hepatoprotection	Increases anti-oxidant activity and oral bioavailability of MPLC SNPs	[56]
MGF treated-AuNP	Human breast cell		0.025 mM to 10 mM		100 μg		Non-toxic	Reduces toxicity	[57]
NSC-MGF nanoconjugate	Blood serum	Ionotropic gelation method	100~200 nm	−30 mV	10 mg/kg	Wistar rat model	Anti-diabetes Anti-hyperlipidemia	Reduces blood glucose level and total plasma cholesterol	[58]

MCNs; Mangiferin-chitosan nanoparticles, MPLC SNPs; phytosomal soft nanoparticles encapsulated with phospholipid complex, MGF-AuNPs; Mangiferin functionalized gold nanoparticulate agent, AuNP; gold nanoparticle, MGF; Mangiferin, NSC; N-succinylated chitosan.

## Data Availability

Not applicable.

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
