# Peer review of "Renoprotective Effects of Mangiferin: Pharmacological Advances and Future Perspectives"

_ijerph, 2022, doi:10.3390/ijerph19031864_

Round 1

Reviewer 1 Report

The study design of each relevant study on kidney diseases in terms of animal models, mangiferin dose, and molecular results. Following that, a summary of the information gathered on the efficacy of mangiferin in kidney diseases is presented. The potential protective mechanisms of mangiferin are also highlighted and discussed in order to close the knowledge gap regarding its use as an alternative treatment for kidney diseases.

  1. The abstract is not clear. Please add the aim and objective of the review.
  2. The MS English needs to be improved. The article's English must be carefully checked for grammatical errors.
  3. In Conclusion, the authors should add the significance of this review, and its potential practical application.

Reviewer 2 Report

The manuscript ijerph-1521059 devoted the actual field of the medical science, namely evaluation of renoprotective effects of Mangiferin and can be interested to the specialists working in this field. The author’s opinion is clear and based on a wide range of recent publications. I am personally impressed by the structure of the article, the systematization of scientific data and the sequence of its presentation. The paper fit the Journal scope and formal requirements. However, it needs minor revision before publication.

To improve the quality and perception of the manuscript I would suggest paying attention to following comments:

  1. It is necessary to present a structural formula of Mangiferin.
  2. It will be good to describe in more detail the sources and methods of obtaining Magniferin, as well as dosage forms based on it.
  3. Moderate English changes required.

My decision is minor revision.

Reviewer 3 Report

This review article is very similar to the accepted review article by Pei Teng Lum et al. “Therapeutic potential of mangiferin against kidney disorders and its mechanism of action: A review” (Accepted 10 November 2021 in the Saudi Journal of Biological Sciences, the file is attached). This fact makes very questionable the possibility of the publication of this review article.

Minor comments:

Line 18: please, rewrite the sentence “However, the existing therapeutic window is too narrow to combat the kidney diseases” – this has incorrect meaning.

Line 21: the sentence “Relevant published literature of the peer-reviewed journals was searched in PubMed and Google Scholar to explore…” has to be deleted.

Line 24: pretreatment – please, correct, because pretreatment has another meaning.

English language has to be corrected throughout the text, because words have another meaning, for example:

Line 26 – appraised,

Line 27 - intercept,

Line 47 – cease,

line 101 – upraises,

line 102 – inaugurates,

line 115 – in subjects,

many etc.

Line 72 - Subtitle 2 Materials and methods has to be deleted.

Line 83 – please, delete “reactive oxygen species” - it was decrypted before.

Line 89 – please, delete “our investigation on several articles”.

Line 104 – please, delete “superoxide dismutase” - it was decrypted before.

Line 123 – correct “Table 1 and 2” to “Tables 1 and 2” or to “Table 1 and Table 2”.

Line 117 – please, decrypt LPS.

Line 130 – please, decrypt NOD- and LRR.

Line 137 – please, correct “alos” to “also”.

Line 142 – please, correct “treatmnet” to “treatment”.

Line 195 – please, correct “suitable candidate drug”.

Subchapter 4 should be deleted or rewritten.

The data in the Table 2 are not discussed, and Table 2 also has to be placed in other part of the article.

The reference number 3 is cited not correctly. Correct:

Matkowski, A.; KuÅ›, P.; Góralska, E.; Dorota Woźniak, D. Mangiferin - a bioactive xanthonoid, not only from mango and not just antioxidant. Mini Rev Med Chem 2013;13(3):439-455.

Most polymer systems for mangiferin delivery are described in the review article:

Morozkina, S.N.; Nhung Vu, T.H.; Generalova, Y.E.; Snetkov, P.P.; Uspenskaya, M.V. Mangiferin as new potential anti-cancer agent and mangiferin-integrated polymer systems-a novel research direction. Biomolecules, 2021, 11(1), 79. https://doi.org/10.3390/biom11010079.

Because Mangiferin possesses the similar mechanisms of pharmacological effects for the treatment of other diseases, some comparisons and conclusions have to be made concerning the particularities of kidney diseases, as well as unique properties of mangiferin in comparison with other medications used for kidney disorders treatment.

Reviewer 4 Report

The authors should consider the followings:

  1. Please enlist and summarize the clinical trials (if any) related to the mangiferin, on the subject matter.
  2. The authors may separately summarize the relevant findings of the aglycone of Mangiferin.
  3. The authors should enlist the possible pharmacological advances to the use of Mangiferin (the pros and cons, per each means)
  4. The authors should define renoprotective effect, in the part of introduction (or abstract).
  5. As per the Figure 1 to 3, please indicate any sources of Figure adoption or modification from (if any) and appropriately cited.
  6. As an review article, please expand the coverage and volumes of the relevant reference literatures.
  7. Combined in a figure, the authors may show the chemical structure of Mangiferin, for their audiences.
  8. The authors should seek English professional for the Use of English in the article.
  9. Table 1 may be displayed as Landscape table for a better presentation.
  10. In Section 2, please indicate the years of collection of the literature covered.

Round 2

Reviewer 3 Report

All my comments have been taken into account.